# Research

medicinal chemistry/biochemistry

resveratrol, synthesis of resveratrol analogues, tubulin, anti-cancer, molecular docking

**Author for correspondence:**
Mingguo Jiang
e-mail: mzxyjiang@163.com

†These authors contributed equally to this work. This article has been edited by the Royal Society of Chemistry, including the commissioning, peer review process and editorial aspects up to the point of acceptance.

# Design, synthesis and biological evaluation of a series of new resveratrol analogues as potential anti-cancer agents

Lifang Yang[1,†], Xuemei Qin[2,†], Hongcun Liu[1], Yanye Wei[2], Hailiang Zhu[3] and Mingguo Jiang[2]

[1]School of Chemistry and Chemical Engineering, Guangxi Key Laboratory of Utilization of Microbial and Botanical Resources, Guangxi Key Laboratory of Chemistry and Engineering of Forest Products, and [2]School of Marine Sciences and Biotechnology, Guangxi Key Laboratory for Polysaccharide Materials and Modifications, Guangxi University for Nationalities, Nanning 530008, People's Republic of China
[3]State Key Laboratory of Pharmaceutical Biotechnology, Nanjing University, Nanjing 210093, People's Republic of China

MJ, 0000-0003-2279-5740

A series of novel resveratrol derivatives were designed, synthesized and evaluated as anti-cancer agents. Most of the compounds showed significant anti-proliferative activities against three human cancer cell lines (HepG2, A549 and Hela). Among these compounds, compound **r** displayed the most potent inhibitory activity and showed low cytotoxic activity. Cell apoptosis and cell cycle assays demonstrated that compound **r** significantly induced apoptosis ($p < 0.001$) and arrested cell cycle at S phase. Immunofluorescence microscopy analysis showed compound **r** disrupted the tubulin network. Docking simulations supported the pharmacological results of compound **r**. It is believed that this work would be very useful for designing a new series of tubulin inhibitors.

## 1. Introduction

Microtubules, composed of α-tubulin and β-tubulin [1,2], control various cellular functions such as maintaining structure of the cell, chromosomal segregation, intracellular transportation and mitosis [3]. In normally growing cells, microtubules are continuously undergoing polymerization and depolymerization and thus are in dynamic equilibrium [4]. This dynamic balance is necessary for cells to complete the normal mitosis process and if the dynamic balance is disrupted, cells undergo cell cycle

**Figure 1.** Design strategy and modification of anti-cancer agent.

arrest and subsequently programmed cell death [4]. Therefore, tubulin has been considered as a promising target for anti-cancer drugs [1].

A large number of natural, synthetic and half synthetic compounds have been identified that target the dynamicity of the tubulin–microtubule system [5,6]. Tubulin inhibitors are divided into three categories on the basis of the microtubule domain targeted by the inhibitor: the paclitaxel-binding domain, the colchicine-binding domain and the vinblastine-binding domain. Vinblastine and paclitaxel, two tubulin inhibitor drugs, have been shown to play important roles in clinical tumour treatment. Paclitaxel, as the first-line anti-cancer drug for clinical treatment, exhibits significant curative effects in non-small cell lung cancer, breast cancer, cervical cancer, ovarian cancer and other cancers [7,8]. However, its clinical efficacy has been limited by side effects and tumour resistance. Therefore, the identification of novel tubulin inhibitors that possess more activity, low side effects and efficacy against multi-drug resistant tumour cells is critical. Natural products are often used as lead compounds in drug discovery owing to their high efficiency, low toxicity and few side effects [9]. Resveratrol (trans-3,4',5-trihydroxy stilbene, figure 1), isolated from grapes, berries and peanuts [10,11], has been the focus of extensive investigation owing to its chemopreventive effect *in vitro* and *in vivo* models [12,13] and shows extremely strong effects on the suppression of tubulin assembly through interaction with the colchicine-binding site of tubulin, resulting in extensive inhibition of cell growth and angiogenesis [14,15]. However, its low oral bioavailability and poor stability [16] has hindered its progress as a potential clinical candidate. *Derris eriocarpa*, a type of the leguminosae family, is used in Zhuang and Dai ethnomedicine to treat various diseases [17,18]. We obtained a series of components from *D. eriocarpa* [19], including a derivative of resveratrol, 3,4,5,4'-tetramethoxy-trans-stilbene (figure 1), which exhibited a striking inhibitory effect on the growth of several cancer cell lines [20,21]. However, pharmaceutical preparation of this resveratrol derivative is difficult owing to its low polarity.

Thus, we designed and synthesized a series of novel small-molecule derivatives of resveratrol by adding a carboxyl group on the carbon–carbon double bond for a better water solubility as a potential anti-cancer agent. Electron-withdrawing and electron-donating groups were chosen as $R^1$–$R^8$ substituents to get a variety of compounds for structure-activity relationship (SAR) studies (figure 1). The synthesized compounds were evaluated for their biological activity against three human cancer cell lines and two normal cells. Most showed significant anti-proliferative activities against the three cancer cell lines and almost no cytotoxic activities against LO2 and Vero cells. Among the compounds, compound **r** demonstrated the most potent anti-proliferative activity against cancer cells ($IC_{50} = 7.49$, 3.77 and 4.79 µM, respectively) and low cytotoxic activities. Cell apoptosis and cell cycle analysis demonstrated that compound **r** induced apoptosis and arrested the cell cycle at S phase. The binding interactions between tubulin and compound **r** were also displayed in docking analysis.

R¹ = –H, –Cl, –CF$_3$, –OCF$_3$, –NO$_2$    R² = –H, –Cl, –OCH$_3$, –CF$_3$    R³ = –H, –Cl, –OCH$_3$, –CF$_3$, –OCF$_3$, –OH    R⁴ = –H, –Cl, –NO$_2$, –CF$_3$

R⁵ = –H, –Cl    R⁶ = –H, –OCH$_3$, –CH$_3$    R⁷ = –H, –OCH$_3$    R⁸ = –H, –OCH$_3$, –CH$_3$

**Scheme 1.** Synthesis of compounds **a–v**. Reagents and conditions: Et$_3$N, (CH$_3$CO)$_2$O, 120°C.

# 2. Results and discussion

## 2.1. Chemistry

The synthesis approach for the 22 resveratrol derivatives **a–v** is depicted in scheme 1. The target compounds were obtained through a Perkin reaction by one step [22–24]. Substituted phenylacetic acid (1 equivalent) was mixed with substituted benzaldehyde (1 equivalent), and then triethylamine (2.5 equivalents) and acetic anhydride (3 equivalents) were added. The mixture was heated at 120°C and stirred overnight. Nitrogen protection was applied during the reaction process to prevent oxidization of aldehyde. After the reaction was completed, 2–3 ml water was added, followed by reflux for 15 min to quench the reaction. The reaction was placed in a water bath at 4°C and 10% K$_2$CO$_3$ solution was added to adjust the reaction to alkaline. The sample was stirred in a water bath at 50°C for 1 h and then HCl was added to adjust the pH to 4–5.

## 2.2. Biological activities assay

### 2.2.1. Anti-proliferation assay

The 22 synthesized resveratrol compounds were evaluated for anti-proliferative activities against three human cancer cell lines, including HepG2, A549 and HeLa cells, using 3-[4,5-dimethylthiazol-2-yl]-2,5 diphenyl tetrazolium bromide (MTT) assays. Paclitaxel, a well-known tubulin inhibitor, was used as a positive control. The IC$_{50}$ values are summarized in table 1. Most of the compounds showed significant anti-proliferative activities against the three human cancer cell lines, and among these compounds, compound **r** exhibited the most potent anti-proliferative activity. Compound **r** potently inhibited the growth of HepG2, A549 and HeLa cancer cells with IC$_{50}$ values of 7.49 µM, 3.77 µM and 4.79 µM, respectively. The anti-proliferative assay results revealed an interesting SAR for the designed resveratrol analogues. Among compounds in which R⁶, R⁷ and R⁸ were all methoxyl groups (compounds **a–n**), compounds **f** (R³ = –OCF$_3$) and **j** (R³ = –CF$_3$), having a strong electron-withdrawing group at R³ position, showed the best inhibitory activities against A549 and HeLa cells; while among compounds in which R⁴ was a strong electron-withdrawing group, compounds **d** (R⁴ = –NO$_2$) and **k** (R⁴ = –NO$_2$) exhibited diminished inhibition against HepG2, A549 and HeLa cells. Among compounds in which R⁶ and R⁸ were H and R⁷ was methoxyl (compounds **o–u**), compounds with an –OCF$_3$ or –CF$_3$ group (compounds **p**, **q** and **r**) showed the most potent inhibitory activities against the three cell lines. Therefore, the inhibitory activity of the compounds depended not only on the 2-position substitution of acrylic acid, but also on the 3-position substitution of acrylic acid.

### 2.2.2. Cytotoxicity test

Vero cells are non-tumorigenic cell lines and are important raw materials as the research and production of biological products. Vero cells show low tumorigenicity and had been approved for use in the production of human virus vaccines. Thus, Vero cells were chosen as a control for cytotoxicity. All compounds were evaluated for their cell toxicity against Vero and normal human hepatocytes cells line (LO2) using MTT assays. As shown in table 2, most of the compounds displayed almost no cytotoxic activities against Vero cells and LO2 cells *in vitro*.

**Table 1.** Structures of compounds (**a**–**v**) and IC$_{50}$ on human cancer cells. (Values are averages of three independent determinations.)

| compound | R$^1$ | R$^2$ | R$^3$ | R$^4$ | R$^5$ | R$^6$ | R$^7$ | R$^8$ | IC$_{50}$ (μM) HepG2 | A549 | HeLa |
|---|---|---|---|---|---|---|---|---|---|---|---|
| a | H | Cl | Cl | H | H | OCH$_3$ | OCH$_3$ | OCH$_3$ | 50 ± 4 | 34 ± 2 | 27 ± 2 |
| b | Cl | H | H | H | Cl | OCH$_3$ | OCH$_3$ | OCH$_3$ | 74 ± 4 | 46 ± 3 | 32 ± 3 |
| c | Cl | H | Cl | H | H | OCH$_3$ | OCH$_3$ | OCH$_3$ | 30 ± 3 | 30 ± 1 | 46 ± 3 |
| d | Cl | H | H | NO$_2$ | H | OCH$_3$ | OCH$_3$ | OCH$_3$ | >100 | >100 | 62 ± 3 |
| e | OCF$_3$ | H | H | H | H | OCH$_3$ | OCH$_3$ | OCH$_3$ | >100 | >100 | 61 ± 3 |
| f | H | H | OCF$_3$ | H | H | OCH$_3$ | OCH$_3$ | OCH$_3$ | 18.7 ± 0.9 | 19 ± 2 | 20 ± 3 |
| g | H | OCF$_3$ | H | H | H | OCH$_3$ | OCH$_3$ | OCH$_3$ | 24 ± 2 | 14 ± 4 | 73 ± 5 |
| h | H | CF$_3$ | H | H | H | OCH$_3$ | OCH$_3$ | OCH$_3$ | 47 ± 4 | 32 ± 5 | 26 ± 4 |
| i | CF$_3$ | H | H | H | H | OCH$_3$ | OCH$_3$ | OCH$_3$ | 42 ± 2 | 32 ± 2 | 41 ± 4 |
| j | H | H | CF$_3$ | H | H | OCH$_3$ | OCH$_3$ | OCH$_3$ | 56.66 ± 0.02 | 18 ± 3 | 19 ± 2 |
| k | Cl | H | H | CF$_3$ | H | OCH$_3$ | OCH$_3$ | OCH$_3$ | 100.0 ± 0.8 | >100 | >100 |
| l | Cl | OCH$_3$ | OCH$_3$ | H | H | OCH$_3$ | OCH$_3$ | OCH$_3$ | >100 | >100 | 89 ± 4 |
| m | NO$_2$ | H | H | Cl | H | OCH$_3$ | OCH$_3$ | OCH$_3$ | >100 | 34 ± 2 | 60 ± 1 |
| n | Cl | Cl | H | H | Cl | OCH$_3$ | OCH$_3$ | OCH$_3$ | 79 ± 4 | 27.2 ± 0.9 | 20 ± 2 |
| o | Cl | H | H | H | Cl | H | OCH$_3$ | H | 48 ± 5 | 21.3 ± 0.7 | 10 ± 2 |
| p | H | H | OCF$_3$ | H | H | H | OCH$_3$ | H | 7.7 ± 0.3 | 14 ± 2 | 5.1 ± 0.8 |
| q | H | CF$_3$ | H | H | H | H | OCH$_3$ | H | 19 ± 2 | 16 ± 3 | 14 ± 1 |
| r | Cl | H | H | CF$_3$ | H | H | OCH$_3$ | H | 7 ± 1 | 3.8 ± 0.3 | 4.8 ± 0.8 |
| s | H | OCH$_3$ | OH | NO$_2$ | H | H | OCH$_3$ | H | 42 ± 4 | 15 ± 4 | 39 ± 2 |
| t | Cl | OCH$_3$ | OCH$_3$ | H | H | H | OCH$_3$ | H | 93 ± 3 | >100 | 24 ± 2 |
| u | Cl | Cl | H | H | Cl | H | OCH$_3$ | H | >100 | >100 | >100 |
| v | H | H | OCH$_3$ | H | H | CH$_3$ | H | CH$_3$ | 100 ± 1 | 61 ± 3 | 51.25 ± 0.01 |
| paclitaxel | | | | | | | | | 0.39 ± 0.03 | 0.17 ± 0.03 | 0.7 ± 0.1 |



| blank (0 µM) | control (20 µM) | low (5 µM) | middle (15 µM) | high (45 µM) |

**Figure 2.** Effect of compound **r** on the microscopic examination of HepG2 cells. Morphology of HepG2 treated with compound **r** (0, 5, 15 and 45 µM) and cisplatin (20 µM) for 24 h. Light microscopy photographs, with 40× objective.

**Table 2.** The median cytotoxic concentration ($CC_{50}$) data of texted compounds. (Values are averages of three independent determinations.)

| compound | $CC_{50}$ (µM) | | compound | $CC_{50}$ (µM) | |
| | Vero | LO2 | | Vero | LO2 |
|---|---|---|---|---|---|
| **a** | 28.2 ± 0.5 | 27 ± 1 | **l** | >100 | 64 ± 2 |
| **b** | >100 | 92 ± 2 | **m** | >100 | >100 |
| **c** | 71 ± 2 | >100 | **n** | 63 ± 2 | 77 ± 1 |
| **d** | >100 | 93 ± 2 | **o** | >100 | >100 |
| **e** | 82 ± 2 | >100 | **p** | 93 ± 3 | 84 ± 2 |
| **f** | >100 | >100 | **q** | >100 | 98 ± 2 |
| **g** | >100 | >100 | **r** | 84 ± 3 | >100 |
| **h** | >100 | 75 ± 4 | **s** | >100 | 83 ± 2 |
| **i** | >100 | >100 | **t** | 85 ± 1 | 87 ± 2 |
| **j** | >100 | >100 | **u** | 38.6 ± 0.7 | 35 ± 1 |
| **k** | 71 ± 2 | 88 ± 2 | **v** | >100 | 96 ± 6 |
| paclitaxel | 98 ± 1 | >100 | | | |

### 2.2.3. Morphologic observation

Morphological observations can directly reflect the effects of drugs on cells. HepG2 cells were cultured with increasing concentrations of compound **r** (0, 5, 15 and 45 µM) for 24 h and observed using a light microscope. The cell morphology was clearly altered with the increase in drug concentration (figure 2). The morphology of the cells treated with compound **r** at 15 and 45 µM presented shrinkage, rounding and detachment from the plate surface, which are representative features of apoptosis. Most of the cells showed a rounded shape upon treatment with 45 µM and cisplatin.

### 2.2.4. Immunofluorescence microscopy assay

Resveratrol was previously reported to hamper microtubule function [25]. To examine whether the anti-proliferative activity of compound **r,** a resveratrol derivative, was related to an interaction with tubulin, we examined β-tubulin in HepG2 cells treated with compound **r** and by immunofluorescence assay. Compound **r** ranging from 5 to 45 µM inhibited β-tubulin for 24 h (figure 3). Compared with the negative control group, cells treated with 5 and 15 µM of compound **r** showed a highly abnormal cytoskeleton; the overall mass of microtubules was reduced and the cells were less well spread. At concentrations of 45 µM, cells were small and showed very little cytoplasm; the microtubules appeared depolymerized. In addition, after incubation with compound **r** (5, 15 and 45 µM), many cells became multinucleated. These phenomena were consistent with those observed with other microtubule-targeted drugs [26]. Compared with cells treated with paclitaxel, cells treated with compound **r** exhibited similar microtubule architecture, including irregular microtubule networks and nuclear localization.

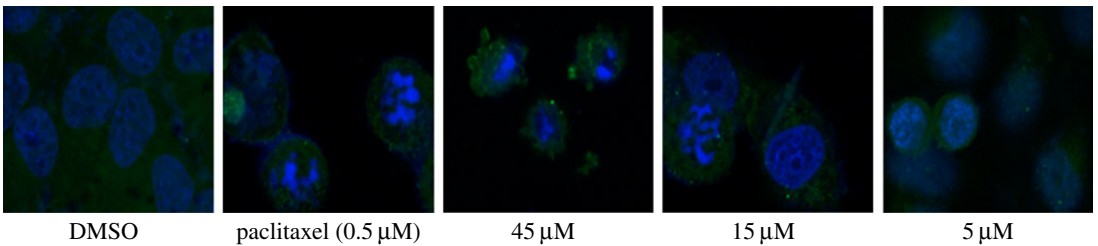

DMSO    paclitaxel (0.5 μM)    45 μM    15 μM    5 μM

**Figure 3.** Compound **r** promoted microtubule depolymerization in HepG2 cells. The HepG2 cells were treated with 0, 5, 15 or 45 μM compound **r** for 24 h, microtubule networks and nuclei were observed by immunofluorescence microscopy assay.

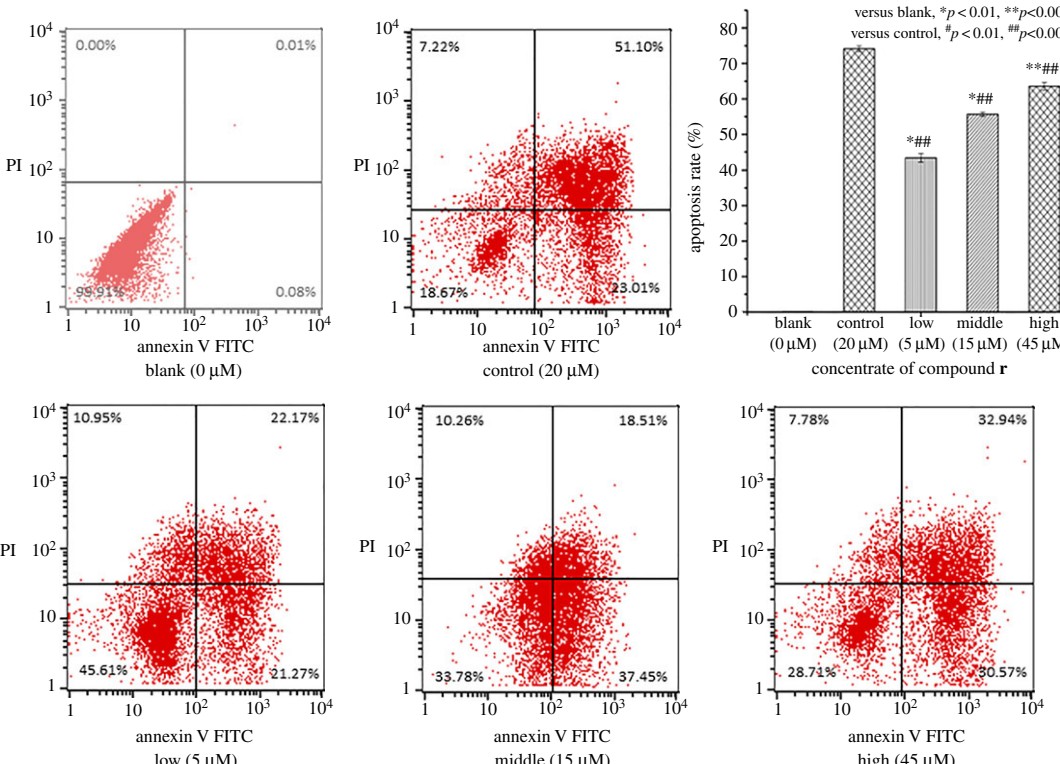

**Figure 4.** Effect of compound **r** on the apoptosis in HepG2 cells. Analysis of compound **r** induced apoptosis in HepG2 cells. The HepG2 cells were treated with 0, 5, 15 or 45 μM by flow cytometry. Cells were stained using propidium iodide (IP) and fluorescein isothiocyanate (FITC). The data manifested the percentage of early apoptotic cells (lower right) or late apoptosis and cell death (upper right) in each quadrant.

### 2.2.5. Apoptotic analysis

To investigate the effect of compound **r** on the apoptosis of HepG2 cells, cells were treated with different concentrations of compound **r** (0, 5, 15 and 45 μM) for 24 h and apoptosis was detected by flow cytometry analysis. The results are shown in figure 4. Increasing concentrations of compound **r** increased the apoptotic cell population in a dose-dependent manner, from 0.09 to 43.44, 55.96 and 63.51%.

### 2.2.6. Cell cycle assay

To evaluate the effects of compound **r** on cell cycle progression of HepG2, cells were treated with compound **r** at different concentrations (0, 5, 15 and 45 μM) for 24 h and then examined by flow cytometry. The results are shown in figure 5. The accumulation of cells in S phase increased with increasing concentration of compound **r**. Approximately $18.36 \pm 1.93\%$, $19.39 \pm 2.06\%$, $29.9 \pm 0.87\%$ and $34.24 \pm 1.00\%$ of cells were arrested in S phase in response to 0, 5, 15 and 45 μM compound **r**,

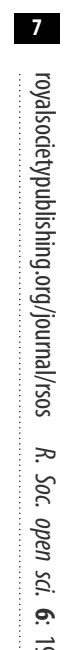

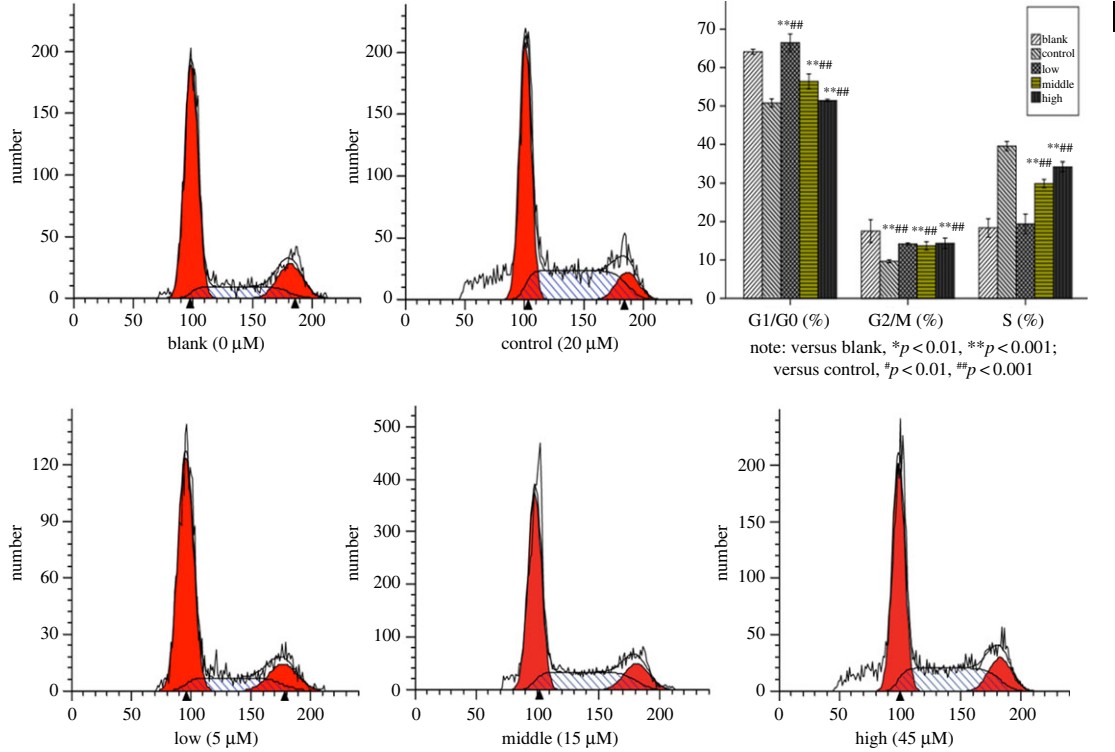

**Figure 5.** Cells treated with 0, 5, 15 and 45 μM compound **r** for 24 h were collected and analysed by flow cytometry. Images were representative of three independent experiments. Values represented the mean ± s.d.; versus blank, *p < 0.01, **p < 0.001; versus control #p < 0.01, ##p < 0.001.

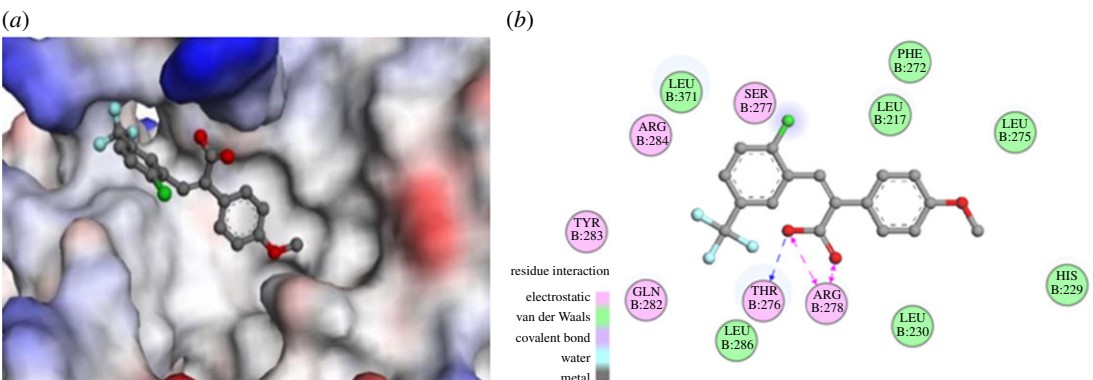

**Figure 6.** (a) Three-dimensional binding model of compound **r** into the active pocket of β-tubulin (PDB ID: 1JFF) and (b) two-dimensional binding mode of compound **r** into the active pocket of β-tubulin.

respectively. In comparison, 39.56 ± 0.97% cells were arrested in S phase after treatment with 20 μM cisplatin for 24 h. These results demonstrated that 45 μM of compound **r** induced a cell cycle arrest to a similar extent as 20 μM of cisplatin.

## 2.3. Molecular docking study with tubulin

To explore the SARs and guide further SAR studies, compound **r** was docked into the paclitaxel-binding site in tubulin to examine the molecular basis for its activity. In addition to the main hydrophobic interaction between compound **r** in the active pocket of tubulin, three direct interactions between compound **r** and two residues (THR276 and ARG278) in tubulin were detected (figure 6), which included one hydrogen bond (blue) and two charge interactions (pink). The corresponding distances and other information of these three interactions are presented in table 3, and the CDOCKER

**Table 3.** The main interactions between compound **r** and two residues (THR276 and ARG278) in the tubulin paclitaxel active pocket.

| interaction type | residue–compound **r** | distance (Å) | angle DHA[a] | angle HAY[a] |
|---|---|---|---|---|
| H-bond | THR276:HG1–**r**:O2 | 2.5 | 141° | 126° |
| charge | ARG278:HG1–**r**:O2 | 2.8 | —[b] | —[b] |
| charge | ARG278:HG1–**r**:O3 | 2.7 | —[b] | —[b] |

[a]D is the hydrogen bond donor heavy atom, H is the explicit donor hydrogen if present, A is the hydrogen bond acceptor atom, Y is any atom attached to the acceptor atom.
[b]None.

interaction energy was calculated to $-34.1123$ kcal mol$^{-1}$, which indicates compound **r** has a certain binding ability with the paclitaxel-binding site in tubulin. The hydrophobic methoxyphenyl of compound **r** was located well into the hydrophobic pocket of tubulin and the hydrophilic carboxyl group formed a hydrogen bond with the active pocket of tubulin. These interactions may provide an experimental explanation for the observation that compound **r** showed the most potent activity in the 22 synthesized compounds. Docking simulations supported the pharmacological results of compound **r**. Whether compound **r** is associated with other unknown targets will be discussed in further studies. The result has implication for further design and development of more potent tubulin inhibitors.

# 3. Experimental methods

## 3.1. General information

All chemicals and reagents were analytical grade. The reagents were purchased and used without further purification. Column chromatography was performed on a 200–300 mesh silica gel column. Melting points were determined on an X-6 microscopic melting point meter (Beijing Tech Instrument Col. Ltd, Beijing, China) and were as read. $^1$H nuclear magnetic resonance (NMR) and $^{13}$C NMR spectra were measured on a Bruker DPX400 spectrometer (Bruker, Germany, 600 MHz, CDCl$_3$ or DMSO). $^1$H NMR spectral data were assigned: chemical shift ($\delta$/ppm), multiplicity (br, broad; m, multiplet; q, quartet; t, triplet; d, doublet; s, singlet), coupling constant ($J$/Hz) and integration. High resolution mass spectroscopy (HRMS) spectra were performed on a MAT95-xp Mass Spectrometer (Thermo Fisher Science, Waltham, MA, USA). MTT assays were read on a Multiskan microplate photometer (Thermo, USA). Immunofluorescence was observed on a Leica TCS SP8 MP fluorescence microscope (Leica Microsystems, Germany). Cell cycle and apoptosis were analysed on a Coulter Epics XL flow cytometer (Becton Dickinson, USA).

## 3.2. Synthesis

General procedure for the preparation of compounds **a–v**. Substituted phenylacetic acid (1 equivalent) and substituted benzaldehyde (1 equivalent) were mixed, then added triethylamine (2.5 equivalents) and acetic anhydride (3 equivalents). The mixture was heated at 120°C and stirred overnight. Nitrogen protection was applied during the reaction process to prevent the aldehyde from oxidization. After the reaction was completed, 2–3 ml water was added, followed by reflux for 15 min to quench the reaction. The reaction was placed in a water bath at 4°C and then 10% K$_2$CO$_3$ solution was added to adjust the reaction to alkaline. The reaction was placed in a water bath at 50°C for 1 h and then HCl was added to adjust the pH to 4–5. The crude product was purified by column chromatography and preparative high-performance liquid chromatography with EtOAc-MeOH gradient to give the desired product.

Compound **a**: 3-(3,4-dichlorophenyl)-2-(3,4,5-trimethoxyphenyl) acrylic acid. Grey powder. Mp. 233–235°C. $^1$H NMR (600 MHz, CDCl$_3$) $\delta$ 3.78 (s, 6H), 3.90 (s, 3H), 6.43 (s, 2H), 6.88 (d, 1H, $J$ = 8.4), 7.26 (t, 2H, $J$ = 6.6), 7.78 (s, 1H). HRMS (ESI): $m/z$ 381.0297 calcd for C$_{18}$H$_{15}$O$_5$Cl$_2$ [M-H]$^-$, found 381.0297.

Compound **b**: 3-(2,6-dichlorophenyl)-2-(3,4,5-trimethoxyphenyl) acrylic acid. Colourless granular. Mp. 217–218°C. $^1$H NMR (600 MHz, CDCl$_3$) $\delta$ 3.66 (s, 6H), 3.80 (s, 3H), 6.41 (s, 2H), 7.11–7.14 (t, 1H, $J$ = 16.4), 7.22–7.24 (d, 2H, $J$ = 7.8), 7.82 (s, 1H). HRMS (ESI): $m/z$ 381.0301 calcd for C$_{18}$H$_{15}$O$_5$Cl$_2$ [M-H]$^-$, found 381.0297.

Compound **c**: 3-(2,4-dichlorophenyl)-2-(3,4,5-trimethoxyphenyl) acrylic acid. Yellow powder. Mp. 228°C. [1]H NMR (600 MHz, CDCl$_3$) $\delta$ 3.74 (s, 6H), 3.87 (s, 3H), 6.39 (s, 2H), 6.78–6.79 (d, 1H, $J$ = 8.4), 6.95–6.96 (t, 1H, $J$ = 10.2), 7.41–7.42 (d, 1H, $J$ = 1.8), 8.10 (s, 1H). [13]C NMR (151 MHz, DMSO-$d_6$): $\delta$ 168.07, 153.08, 136.91, 134.89, 134.33, 134.20, 132.78, 132.11, 130.58, 129.32, 127.48, 107.54, 60.54, 56.28. HRMS (ESI): $m/z$ 381.0297 calcd for C$_{18}$H$_{15}$O$_5$Cl$_2$ [M-H]$^-$, found 381.0295.

Compound **d**: 3-(2-chloro-5-nitrophenyl)-2-(3,4,5-trimethoxyphenyl) acrylic acid. Yellow floccule. Mp. 247–248°C. [1]H NMR (600 MHz, CDCl$_3$) $\delta$ 3.91 (s, 3H), 3.94 (s, 6H), 6.95 (s, 2H), 7.50 (d, 1H, $J$ = 9.0), 7.88 (s, 1H), 8.40 (d, 1H), 8.51 (s, 1H). HRMS (ESI): $m/z$ 392.0537 calcd for C$_{18}$H$_{15}$NO$_7$Cl [M-H]$^-$, found 392.0537.

Compound **e**: 3-(2-(trifluoromethoxy) phenyl)-2-(3,4,5-trimethoxyphenyl) acrylic acid. White acicular crystal. Mp. 164°C. [1]H NMR (600 MHz, DMSO-$d_6$) $\delta$ 3.60 (s, 6H), 3.66 (s, 3H), 6.39 (s, 2H), 6.94 (d, 1H, $J$ = 7.8), 7.18–7.19 (m, 1H), 7.39–7.40 (m, 2H), 7.79 (s, 1H). HRMS (ESI): $m/z$ 397.0899 calcd for C$_{19}$H$_{16}$O$_6$F$_3$ [M-H]$^-$, found 397.0900.

Compound **f**: 3-(4-(trifluoromethoxy) phenyl)-2-(3, 4, 5-trimethoxyphenyl) acrylic acid. White powder. Mp. 208°C. [1]H NMR (600 MHz, DMSO-$d_6$) $\delta$ 3.65 (s, 6H), 3.70 (s, 3H), 6.45 (s, 2H), 7.24–7.24 (m, 4H), 7.73 (s, 1H). HRMS (ESI): $m/z$ 397.0899 calcd for C$_{19}$H$_{16}$O$_6$F$_3$ [M-H]$^-$, found 397.0899.

Compound **g**: 3-(3-(trifluoromethoxy) phenyl)-2-(3, 4, 5-trimethoxyphenyl) acrylic acid. White acicular crystal. Mp.156–157°C. [1]H NMR (600 MHz, CDCl$_3$) $\delta$ 3.77 (s, 6H), 3.89 (s, 3H), 6.44 (s, 2H), 6.91 (s, 1H), 7.10–7.12 (m, 2H), 7.26–7.29 (m, 1H), 7.88 (s, 1H). [13]C NMR (151 MHz, pyridine) $\delta$ 154.08, 149.24, 140.63, 138.41, 136.39, 130.05, 129.68, 122.58, 122.36, 106.61, 61.20, 56.35, 30.00. HRMS (ESI): $m/z$ 397.0899 calcd for C$_{19}$H$_{16}$O$_6$F$_3$ [M-H]$^-$, found 397.0903.

Compound **h**: 3-(3-(trifluoromethyl) phenyl)-2-(3,4,5-trimethoxyphenyl) acrylic acid. Granular. Mp. 162–164°C. [1]H NMR (600 MHz, CDCl$_3$) $\delta$ 3.76 (s, 6H), 3.89 (s, 3H), 6.44 (s, 2H), 7.26–7.28 (m, 1H), 7.32–7.36 (m, 2H), 7.50 (d, 1H, $J$ = 7.8), 7.93 (s, 1H). HRMS (ESI): $m/z$ 381.0950 calcd for C$_{19}$H$_{16}$O$_5$F$_3$ [M-H]$^-$, found 381.0952.

Compound **i**: 3-(2-(trifluoromethyl) phenyl)-2-(3, 4, 5-trimethoxyphenyl) acrylic acid. Colourless granular. Mp. 160–162°C. [1]H NMR (600 MHz, CDCl$_3$) $\delta$ 3.66 (s, 6H), 3.82 (s, 3H), 6.36 (s, 2H), 6.94 (d, 1H, $J$ = 7.8), 7.24–7.26 (t, 1H, $J$ = 15.3), 7.31–7.34 (m, 1H), 7.66–7.68 (d, 1H), 8.22 (s, 1H). HRMS (ESI): $m/z$ 381.0950 calcd for C$_{19}$H$_{16}$O$_5$F$_3$ [M-H]$^-$, found 381.0946.

Compound **j**: 3-(4-(trifluoromethyl) phenyl)-2-(3, 4, 5-trimethoxyphenyl) acrylic acid. White powder. Mp.259°C. [1]H NMR (600 MHz, CDCl$_3$) $\delta$ 3.58 (s, 6H), 3.82 (s, 3H), 6.30 (s, 2H), 7.06 (s, 2H), 7.33 (s, 2H), 7.82 (s, 1H). HRMS (ESI): $m/z$ 381.0950 calcd for C$_{19}$H$_{16}$O$_5$F$_3$ [M-H]$^-$, found 381.0959.

Compound **k**: 3-(2-chloro-5-(trifluoromethyl) phenyl)-2-(3,4,5-trimethoxyphenyl) acrylic acid. Yellow powder. Mp.258°C. [1]H NMR (600 MHz, DMSO) $\delta$ 3.71 (s, 3H), 3.82 (s, 6H), 6.80 (s, 2H), 7.15 (s, 1H), 7.74–7.80 (m, 3H). HRMS (ESI): $m/z$ 415.0560 calcd for C$_{19}$H$_{15}$O$_5$F$_3$Cl [M-H]$^-$, found 415.0565.

Compound **l**: 3-(2-chloro-3,4-dimethoxyphenyl)-2-(3,4,5-trimethoxyphenyl) acrylic acid. Yellow granules. Mp.171–173°C. [1]H NMR (600 MHz, CDCl$_3$) $\delta$ 3.75 (s, 6H), 3.82 (s, 3H), 3.86 (s, 3H), 3.87 (s, 3H), 6.42 (s, 2H), 6.53 (d, 2H, $J$ = 9.0), 6.61 (d, 1H, $J$ = 9.0), 7.21 (s, 1H). HRMS (ESI): $m/z$ 407.0898 calcd for C$_{20}$H$_{20}$O$_7$Cl [M-H]$^-$, found 407.0890.

Compound **m**: 3-(5-chloro-2-nitrophenyl)-2-(3,4,5-trimethoxyphenyl) acrylic acid. White granules. Mp. 167–169°C. [1]H NMR (600 MHz, CDCl$_3$) $\delta$ 3.86 (s, 3H), 3.88 (s, 6H), 6.91 (s, 1H), 6.94 (s, 1H), 7.30 (d, 1H, $J$ = 9.0), 7.99 (d, 1H, $J$ = 9.0), 8.10–8.10 (s, 1H). HRMS (ESI): $m/z$ 392.0537 calcd for C$_{18}$H$_{15}$NO$_7$Cl [M-H]$^-$, found 392.0533.

Compound **n**: 3-(2,3,6-trichlorophenyl)-2-(3,4,5-trimethoxyphenyl) acrylic acid. Yellow granules. Mp. 203–204°C. [1]H NMR (600 MHz, CDCl$_3$) $\delta$ 3.68 (s, 6H), 3.81 (s, 3H), 6.40 (s, 2H), 7.17 (d, 1H, $J$ = 9.0), 7.30 (d, 1H, $J$ = 9.0), 7.78 (s, 1H). HRMS (ESI): $m/z$ 414.9907 calcd for C$_{18}$H$_{14}$O$_5$Cl$_3$ [M-H]$^-$, found 414.9901.

Compound **o**: 3-(2,6-dichlorophenyl)-2-(4-methoxyphenyl) acrylic acid. Colourless powder. Mp. 319–320°C. [1]H NMR (600 MHz, CDCl$_3$) $\delta$ 3.64 (s, 3H), 6.60–6.62 (d, 2H), 6.94 (d, 2H, $J$ = 8.0), 7.09 (s, 1H), 7.17–7.18 (m, 1H), 7.31 (d, 2H, $J$ = 7.8). [13]C NMR (151 MHz, DMSO-$d_6$) $\delta$ 170.29, 158.02, 146.37, 136.66, 134.07, 131.65, 130.71, 129.46, 128.31, 126.29, 112.51, 55.19. HRMS (ESI): $m/z$ 321.0085 calcd for C$_{16}$H$_{11}$O$_3$Cl$_2$ [M-H]$^-$, found 321.0084.

Compound **p**: 2-(4-methoxyphenyl)-3-(4-(trifluoromethoxy) phenyl) acrylic acid. White granular. Mp. 159–160°C. [1]H NMR (600 MHz, CDCl$_3$) $\delta$ 3.84 (s, 3H), 6.92–6.93 (d, 2H, $J$ = 7.2), 7.02–7.03 (d, 2H, $J$ = 8.2), 7.12–7.16 (m, 4H), 7.87 (s, 1H). [13]C NMR (151 MHz, CDCl$_3$) $\delta$ 171.96, 159.32, 149.22, 139.29, 133.32, 132.06, 130.93, 127.26, 120.32, 114.20, 55.17, 40.23. HRMS (ESI): $m/z$ 337.0688 calcd for C$_{17}$H$_{12}$O$_4$F$_3$ [M-H]$^-$, found 337.0692.

Compound **q**: 2-(4-methoxyphenyl)-3-(3-(trifluoromethyl) phenyl) acrylic acid. White powder. Mp. 192–194°C. [1]H NMR (600 MHz, DMSO-$d_6$) $\delta$ 3.84 (s, 3H), 6.92 (d, 3H, $J$ = 8.4), 7.15 (d, 2H, $J$ = 8.4),

7.23–7.30 (m, 2H), 7.47 (d, 2H, $J = 7.8$), 7.91 (s, 1H). HRMS (ESI): $m/z$ 321.0739 calcd for $C_{17}H_{12}O_3F_3$ [M-H]$^-$, found 321.0739.

Compound **r**: 3-(2-chloro-5-(trifluoromethyl) phenyl)-2-(4-methoxyphenyl) acrylic acid. White powder. Mp.140–142°C. $^1$H NMR (600 MHz, CDCl$_3$) $\delta$ 3.82 (s, 3H), 6.85 (d, 3H, $J = 8.4$), 7.37 (d, 1H, $J = 7.2$), 7.42–7.44 (m, 3H), 8.03 (s, 1H). $^{13}$C NMR (151 MHz, CDCl$_3$) $\delta$ 171.17, 160.12, 138.41, 137.45, 135.67, 129.91, 129.07, 128.85, 128.80, 128.45, 126.34, 126.31, 125.51, 125.36, 114.07, 55.36, 39.93. HRMS (ESI): $m/z$ 355.0349 calcd for $C_{17}H_{11}O_3F_3Cl$ [M-H]$^-$, found 355.0345.

Compound **s**: 3-(4-hydroxy-3-methoxy-5-nitrophenyl)-2-(4-methoxyphenyl) acrylic acid. Red powder. Mp. 151–152°C. $^1$H NMR (600 MHz, DMSO-$d_6$) $\delta$ 3.77 (s, 3H), 3.78 (s, 3H), 6.74 (s, 1H), 6.95 (d, 2H, $J = 9$), 7.21 (s, 1H), 7.41 (d, 2H, $J = 8.4$), 7.62 (s, 1H). $^{13}$C NMR (151 MHz, DMSO-$d_6$) $\delta$ 171.64, 159.28, 127.33, 114.46, 56.41, 55.61. HRMS (ESI): $m/z$ 344.0770 calcd for $C_{17}H_{14}NO_7$ [M-H]$^-$, found 344.0769.

Compound **t**: 3-(2-chloro-3,4-dimethoxyphenyl)-2-(4-methoxyphenyl) acrylic acid. Yellow granules. Mp. 202°C. $^1$H NMR (600 MHz, CDCl$_3$) $\delta$ 3.80 (s, 3H), 3.82 (s, 3H), 3.86 (s, 3H), 6.50 (d, 1H, $J = 9.0$), 6.59 (d, 1H, $J = 9.0$), 6.86 (d, 2H, $J = 8.4$), 7.14 (d, 2H, $= 9.0$), 8.14 (s, 1H). HRMS (ESI): $m/z$ 347.0686 calcd for $C_{18}H_{16}O_5Cl$ [M-H]$^-$, found 347.0685.

Compound **u**: 2-(4-methoxyphenyl)-3-(2,3,6-trichlorophenyl) acrylic acid. Brown powder. Mp. 174°C. $^1$H NMR (600 MHz, CDCl$_3$) $\delta$ 3.76 (s, 3H), 6.74 (d, 2H, $J = 8.4$), 7.06 (d, 2H, $J = 8.4$), 7.15 (d, 1H, $J = 8.0$), 7.29 (d, 1H, $J = 8.4$), 7.77 (s, 1H). HRMS (ESI): $m/z$ 354.9696 calcd for $C_{16}H_{10}O_3Cl_3$ [M-H]$^-$, found 354.9688.

Compound **v**: 2-(3,5-dimethylphenyl)-3-(4-methoxyphenyl) acrylic acid. White powder. Mp. 198–200°C. $^1$H NMR (600 MHz, DMSO-$d_6$) $\delta$ 2.25 (s, 6H), 3.70 (s, 3H): 6.76–6.77 (m, 4H), 6.99–7.03 (m, 3H), 7.66 (s, 1H). $^{13}$C NMR (151 MHz, CDCl$_3$) $\delta$ 173.13, 160.52, 141.56, 138.32, 135.60, 132.77, 129.64, 129.27, 127.17, 127.07, 113.70, 55.23, 21.35. HRMS (ESI): $m/z$ 281.1178 calcd for $C_{18}H_{17}O_3$ [M-H]$^-$, found 281.1179.

## 3.3. Biological assays

### 3.3.1. Anti-proliferative assay and cell proliferation toxicities assay

The anti-proliferative and cytotoxicity activity was determined using MTT assay. Human cervical carcinoma HeLa cells, human hepatocellular carcinoma HepG2 cells, human lung carcinoma A549 cells were representative in the anti-proliferation assay, which were obtained from the Shanghai Institute of Cell Biology (Shanghai, China). Normal African green monkey kidney Vero cells were a gift from Dr Xiumiao He at Guangxi University for Nationalities. Normal human hepatocytes LO2 cells were a gift from Dr Xing Lin at Guangxi Medical University. The HepG2, Hela, LO2 and Vero cells were cultured in Dulbecco's modified Eagle's medium (DMEM, Gibco) and the A549 cells were cultured in RPMI1640 (Gibco), both media were supplemented with 10% fetal calf serum and antibiotics (100 IU ml$^{-1}$ penicillin, 100 IU ml$^{-1}$ streptomycin; Amresco).

Cells were seeded in 96-well microtiter plates at a density of $1 \times 10^4$ cells well$^{-1}$ and incubated with different concentrations of compound (6.25, 12.50, 25.00, 50.00 to 100.00 µM), cisplatin and paclitaxel in RPMI-1640 or DMEM at 37°C with 5% CO$_2$. After 24 h, cell morphology was observed using a light microscope, and then 20 µl of MTT solution (5 mg ml$^{-1}$) was added to the well and incubated for 4 h, then cell survival was determined by measuring the optical density (OD) absorbance at a wavelength of 570 nm using an ELISA microplate reader. The inhibition rate (IR) following each compound treatment was determined by the following equation: IR (%) = (1 − ODexp/ODcon) × 100%, where ODexp and ODcon are the OD of the treated and untreated cells, respectively. Cisplatin and paclitaxel were used as positive control drugs.

### 3.3.2. Immunofluorescence assay

HepG2 cells were cultured in DMEM supplemented with 10% fetal bovine serum on coverslips in 6-well plates and allowed to adhere for at least 12 h. Cells were then incubated in the presence or absence of compound **r** (0, 5, 15, 45 µM) for 24 h. Control cells were incubated with vehicle alone. Cells were fixed in 4% paraformaldehyde (15 min, 37°C), washed with phosphate buffered saline (PBS) three times and permeabilized with 1% Triton X-100 in PBS for 10 min. Cells were then incubated with anti-β-tubulin antibody (1 : 200) (Solarbio, China) for 1 h, followed by incubation with rhodamine-conjugated secondary antibody (1 : 500) for 1 h. Cell nuclei were stained with 4,6-diamidino-2-phenylindole (DAPI) and then images were obtained with a fluorescence microscope equipped with a motorized image capture system. Paclitaxel were used as positive control drugs.

### 3.3.3. Evaluation of apoptosis

HepG2 cells seeded in 6-well plates were treated with compound **r** (0, 5, 15, 45 µM) or cisplatin (20 µM) for 24 h. Cells were stained using the Annexin V-FITC/PI apoptosis detection kit (BD Biosciences, USA) in accordance with the manufacturer's instructions. The stained cells were detected via a flow cytometer.

### 3.3.4. Analysis of the cell cycle

Cells were treated with compound **r** (0, 5, 15, 45 µM) or 20 µM of cisplatin as the control for 24 h. Cells were collected by centrifugation (1000 rpm, 5 min), washed with PBS and fixed with 70% ice-cold ethanol overnight at 4°C. The cell cycle was examined on a flow cytometer using a cell cycle analysis kit (Beyotime Biotechnologyin, China) in accordance with the manufacturer's instructions.

## 3.4. Docking study

The molecular docking of compound **r** into the three-dimensional β-tubulin complex structure (PDB ID: 1JFF; downloaded from Protein Date Bank, http://www.rcsb.org/pdb/home/home.do) was performed using CDOCK. The docking study was performed to position compound **r** into the paclitaxel-binding site to determine the probable binding conformation. The depicted CDOCKER INTERATION ENERGY was used as the evaluation criteria.

# 4. Conclusion

A series of novel resveratrol derivatives were designed, synthesized and biological activities were evaluated. The results showed that most of the compounds displayed good anti-proliferative activity against HepG2 cells, A549 cells and HeLa cells. Among the series of novel resveratrol derivatives, compound **r** showed the most potent anti-proliferative activity in HepG2 cells, A549 cells and HeLa cells with $IC_{50}$ values of 7.49, 3.77 and 4.79 µM, respectively. Most of the tested compounds displayed almost no cytotoxicity *in vitro* against Vero cells compared with the positive control paclitaxel. Compound **r** induced apoptosis by microtubule polymerization obstruction resulting in arrest of the cell cycle at the S phase. The docking study showed that compound **r** could be nicely bound to the paclitaxel-binding pocket of tubulin. These findings highlighted the potential of compound **r** as new anti-cancer agents targeting tubulin through arresting the cell cycle at the S phase, inducing apoptotic cell death and blocking tubulin polymerization.

A series of 22 novel resveratrol derivatives were designed, synthesized and biological activities were evaluated. Most of the compounds displayed good anti-proliferative activity against HepG2, A549 and HeLa cells. Among the novel resveratrol derivatives, compound **r** showed the most potent anti-proliferative activity in HepG2, A549 and HeLa cells with $IC_{50}$ values of 7.49, 3.77 and 4.79 µM, respectively. Most of the tested compounds displayed almost no cytotoxicity *in vitro* against Vero cells and LO2 cells compared with the positive control paclitaxel. Compound **r** induced apoptosis by disruption of microtubule polymerization, resulting in cell cycle arrest at S phase. The docking study showed that compound **r** bound to the paclitaxel-binding pocket of tubulin. These findings highlight the potential of compound **r** as a new anti-cancer agent targeting tubulin through blocking tubulin polymerization, arresting the cell cycle at S phase and inducing apoptotic cell death.

Data accessibility. The supporting data for this article have been uploaded as part of the electronic supplementary material. This includes various NMR and HRMS spectra.

Authors' contributions. M.J. contributed to the conception of the study. L.Y. and X.Q. supervised all the experiments, drafted the manuscript and edited the final version of the manuscript. H.L. and Y.W. contributed to synthesis and the activity tests. H.Z. performed the molecular docking work.

Competing interests. We declare we have no competing interests.

Funding. This work was supported by the National Natural Science Foundation of China (81560713), Guangxi Provincial Natural Science Foundation (2016GXNSFAA380075), major research project of Guangxi for science and technology (AA18242026), specific research project of Guangxi for research bases and talents (AD18126005), the middle-aged and young teachers' basic ability promotion project of Guangxi (2017KY0165).

Acknowledgements. We are thankful to Dongdong Li and Yushun Yang for the molecular docking work. We thank Liwen Bianji, Edanz Editing China (www.liwenbianji.cn/ac), for editing the English text of a draft of this manuscript.

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
