## [Reviewer comments · Royal Society Open Science]

Review History

RSOS-190125.R0 (Original submission)

Review form: Reviewer 1

Is the manuscript scientifically sound in its present form?

No

Are the interpretations and conclusions justified by the results?

No

Is the language acceptable?

No

Is it clear how to access all supporting data?

No

Do you have any ethical concerns with this paper?

No

Have you any concerns about statistical analyses in this paper?

No

Recommendation?

Reject

Comments to the Author(s)

The authors present the synthesis and biological activity assays of a series of novel resveratrol analogues. The study identifies compound r as the most potent. However it is considerably less potent than Taxol. There is not data provided as to the purity of these compounds, no HPLC, no NMR traces. This, combined with the issues listed below prevent me from supporting publication at this time.

Major issues:

No NMR, No HPLC data. No purity information Is the biological activity attributable to the compound or an impurity?

In the morphological observations it is not clear why the compounds were combined with cisplatin. It is not stated in the text that cisplatin was used but it was in the figure legend.

The NMR data is assigned to three significant figures. How accurate is this?

Minor issue:

The manuscript needs further proof reading to fix spelling and grammar issues.

For example the use of N2 for protection is not a clear description of what I assume to mean inert atmosphere.

'Lead compounds are often used as lead compounds for drug discovery' This is not correct. They are often used as hit compounds but never leads.

Figure 5, the graph is too small to read the labels.

Review form: Reviewer 2

Is the manuscript scientifically sound in its present form?

Yes

Are the interpretations and conclusions justified by the results?

No

Is the language acceptable?

No

Is it clear how to access all supporting data?

Yes

Do you have any ethical concerns with this paper?

No

Have you any concerns about statistical analyses in this paper?

Yes

Recommendation?

Major revision is needed (please make suggestions in comments)

Comments to the Author(s)

In this manuscript, Yang et. al. reported the synthesis of 22 resveratrol derivatives and characterized their anti-cancer activities. The synthesis is straightforward by coupling two scaffolds under a single condition. The resulting compounds were examined with MTT assay and, with the compound "r", for its effects on cellular morphology, tubulin formation, apoptosis and cell cycle. The authors claimed tubulin as a potent target of the compound "r" based on the evidence of molecular docking. While the exact mechanism of action for the compound remains unclear, the decent effort has been made to prepare these compounds and to characterize the most promising one. I suggest publishing this work after addressing the following concerns:

(1) There are numerous grammar errors via the text. A few examples are: Line 10, missing "of" between "series" and "new"; Line 31, "agent" should be "agents"; Line 33, "cell cycle" should be "cell cycle assays"; Line 34, should be "... arrested cell cycle at S-phase; Line 35, the sentence "... compound r anti-cancer drug targeting tubulin..." is not clear; Line 35, "were" should be "was"; Line 36, "the tubulin active pocket" should be "the active pocket of tubulin"; Line 36, "probable" should be "possible". There are more via the text. I suggest authors to seek the help of an editor company or a native speaker to address these issues.

(2) For Scheme 1, the substituents of R1-R8 should be listed given that there are only 22 compounds.

(3) For the section 2.1, the authors should provide clear rationale why this scaffold was chosen. What is the reason for selecting R1-R8 substituents in the context of prior knowledge of the family of compounds.

(4) For 2.2.1, the choices of cell lines should be rationalized in the context of known biology.

(5) Why was Vero cells chosen as control?

(6) Need the rationale the experiment 2.2.3. Why does the morphology matter?

(7) There is also a limitation of docking. In addition, even the compound r indeed targets tubulin. The authors still could not conclude that the biology of the compound is associated with tubulin interaction but not other unknown targets. The limitation should be clearly spoken out.

(8) There is no error range of IC50 in Table 1 and CC50 in Table. In addition, the accuracy could not reach the current digits. These data should be represented including the correct accuracy and errors.

(9) For Table 3, the distances and angles with such accuracy could not be obtained with the current modeling methods. The data need to be presented in the correct format.

Decision letter (RSOS-190125.R0)

24-Apr-2019

Dear Dr Jiang:

Title: Design, synthesis and biological evaluation of a series new resveratrol analogues as potential anti-cancer agent

Manuscript ID: RSOS-190125

The editor assigned to your manuscript has now received comments from reviewers. We would like you to revise your paper in accordance with the referee and Subject Editor suggestions which can be found below (not including confidential reports to the Editor). Please note this decision does not guarantee eventual acceptance.

Please submit your revised paper before 17-May-2019. Please note that the revision deadline will expire at 00.00am on this date. If we do not hear from you within this time then it will be assumed that the paper has been withdrawn. In exceptional circumstances, extensions may be possible if agreed with the Editorial Office in advance. We do not allow multiple rounds of revision so we urge you to make every effort to fully address all of the comments at this stage. If deemed necessary by the Editors, your manuscript will be sent back to one or more of the original reviewers for assessment. If the original reviewers are not available we may invite new reviewers.

Please also include the following statements alongside the other end statements. As we cannot publish your manuscript without these end statements included, if you feel that a given heading is not relevant to your paper, please nevertheless include the heading and explicitly state that it is not relevant to your work.

- Ethics statement

Please clarify whether you received ethical approval from a local ethics committee to carry out your study. If so please include details of this, including the name of the committee that gave consent in a Research Ethics section after your main text. Please also clarify whether you received informed consent for the participants to participate in the study and state this in your Research Ethics section.

OR

Please clarify whether you obtained the necessary licences and approvals from your institutional animal ethics committee before conducting your research. Please provide details of these licences and approvals in an Animal Ethics section after your main text.

OR

Please clarify whether you obtained the appropriate permissions and licences to conduct the fieldwork detailed in your study. Please provide details of these in your methods section.

On behalf of the Subject Editor Professor Anthony Stace and the Associate Editor Dr Andrew Harned.

RSC Associate Editor:

Comments to the Author:

The reviewers have raised several relevant concerns with this work and the way it is presented. The authors should carefully consider these comments and submit a revised manuscript that addresses these concerns, particularly with respect to compound purity. Any subsequent review will not proceed without evidence of compound purity.

RSC Subject Editor:

Comments to the Author:

(There are no comments.)

Reviewers' Comments to Author:

Reviewer: 1

Comments to the Author(s)

The authors present the synthesis and biological activity assays of a series of novel resveratrol analogues. The study identifies compound r as the most potent. However it is considerably less potent than Taxol. There is not data provided as to the purity of these compounds, no HPLC, no NMR traces. This, combined with the issues listed below prevent me from supporting publication at this time.

Major issues:

No NMR, No HPLC data. No purity information Is the biological activity attributable to the compound or an impurity?

In the morphological observations it is not clear why the compounds were combined with cisplatin. It is not stated in the text that cisplatin was used but it was in the figure legend.

The NMR data is assigned to three significant figures. How accurate is this?

Minor issue:

The manuscript needs further proof reading to fix spelling and grammar issues.

For example the use of N₂ for protection is not a clear description of what I assume to mean inert atmosphere.

'Lead compounds are often used as lead compounds for drug discovery' This is not correct. They are often used as hit compounds but never leads.

Figure 5, the graph is too small to read the labels.

Reviewer: 2

Comments to the Author(s)

In this manuscript, Yang et. al. reported the synthesis of 22 resveratrol derivatives and characterized their anti-cancer activities. The synthesis is straightforward by coupling two scaffolds under a single condition. The resulting compounds were examined with MTT assay and, with the compound "r", for its effects on cellular morphology, tubulin formation, apoptosis and cell cycle. The authors claimed tubulin as a potent target of the compound "r" based on the evidence of molecular docking. While the exact mechanism of action for the compound remains unclear, the decent effort has been made to prepare these compounds and to characterize the most promising one. I suggest publishing this work after addressing the following concerns:

(1) There are numerous grammar errors via the text. A few examples are: Line 10, missing "of" between "series" and "new"; Line 31, "agent" should be "agents"; Line 33, "cell cycle" should be "cell cycle assays"; Line 34, should be "... arrested cell cycle at S-phase; Line 35, the sentence "... compound r anti-cancer drug targeting tubulin..." is not clear; Line 35, "were" should be "was"; Line 36, "the tubulin active pocket" should be "the active pocket of tubulin"; Line 36, "probable" should be "possible". There are more via the text. I suggest authors to seek the help of an editor company or a native speaker to address these issues.

(2) For Scheme 1, the substituents of R1-R8 should be listed given that there are only 22 compounds.

(3) For the section 2.1, the authors should provide clear rationale why this scaffold was chosen. What is the reason for selecting R1-R8 substituents in the context of prior knowledge of the family of compounds.

(4) For 2.2.1, the choices of cell lines should be rationalized in the context of known biology.

(5) Why was Vero cells chosen as control?

(6) Need the rationale the experiment 2.2.3. Why does the morphology matter?

(7) There is also a limitation of docking. In addition, even the compound r indeed targets tubulin. The authors still could not conclude that the biology of the compound is associated with tubulin interaction but not other unknown targets. The limitation should be clearly spoken out.

(8) There is no error range of IC50 in Table 1 and CC50 in Table. In addition, the accuracy could not reach the current digits. These data should be represented including the correct accuracy and errors.

(9) For Table 3, the distances and angles with such accuracy could not be obtained with the current modeling methods. The data need to be presented in the correct format.

Author's Response to Decision Letter for (RSOS-190125.R0)

See Appendix A.

RSOS-190125.R1 (Revision)

Review form: Reviewer 1

Is the manuscript scientifically sound in its present form?

Yes

Are the interpretations and conclusions justified by the results?

Yes

Is the language acceptable?

Yes

Is it clear how to access all supporting data?

Yes

Do you have any ethical concerns with this paper?

No

Have you any concerns about statistical analyses in this paper?

No

Recommendation?

Accept with minor revision (please list in comments)

Comments to the Author(s)

The authors have addressed my revisions well. I recommend publication upon correction of the revision noted below.

Figure 1. The bond angle of the COOH on the double bond needs to be corrected.

Review form: Reviewer 2

Is the manuscript scientifically sound in its present form?

No

Are the interpretations and conclusions justified by the results?

Yes

Is the language acceptable?

Yes

Is it clear how to access all supporting data?

Not Applicable

Do you have any ethical concerns with this paper?

No

Have you any concerns about statistical analyses in this paper?

Yes

Recommendation?

Accept with minor revision (please list in comments)

Comments to the Author(s)

The revised manuscript addressed most of my previous concerns. However, the authors should address the following before consideration of publishing this work. In particular, the answers to

many questions should be included in the text to increase clarification rather than simply being used to answer the reviewer's questions.

For the Q3, the authors' answer is relevant. The following paragraph should be embedded within the text somewhere.

"Resveratrol shows extremely strong effects on suppression of tubulin assembly through interaction with the colchicine binding site of tubulin, resulting in extensive inhibition of cell growth and angiogenesis. However, it is difficult to be made pharmaceutical preparation due to low polarity. Thus, we designed and synthesized a series of novel small-molecule derivatives of resveratrol by adding a carboxyl group on the carbon-carbon double bond for a better water solubility as potential anti-cancer agent. Electron-withdrawing and electron-donating group were chosen as R1-R8 substituents to get variety compounds for SAR studies."

For the Q4, I don't feel that the authors understood the meaning of "rationalize", although they answered the question indirectly by saying "A549, Hela and HepG2 cell lines are representative". The detailed description of these cell lines should be moved to the experimental session. "A549, Hela and HepG2 cell lines are representative" should be included in the main text to tell readers the reason to choose these cell lines.

For the Q5, thanks for providing the answers. This answer should be included in the text rather than just letting me know.

For the Q6, Q7, the answers should be included in the text.

For the Q8, the format to present "mean +/- error" was not correct. For instance, "50.21±3.85" should be "50±4"; "28.17±0.53" should be "28.2±0.5" for statistical reasons. The changes should be made via the whole manuscript.

For the Q9, the current modeling could not present the accuracy to 0.01 Å and 0.01 degree. For instance, "2.48" is better shown as "2.4"; "114.07" is better shown as "141". These numbers should be updated.

Decision letter (RSOS-190125.R1)

25-Jun-2019

Dear Dr Jiang:

Title: Design, synthesis and biological evaluation of a series of new resveratrol analogues as potential anti-cancer agents
Manuscript ID: RSOS-190125.R1

Thank you for submitting the above manuscript to Royal Society Open Science. On behalf of the Editors and the Royal Society of Chemistry, I am pleased to inform you that your manuscript will be accepted for publication in Royal Society Open Science subject to minor revision in accordance with the referee suggestions. Please find the reviewers' comments at the end of this email.

The reviewers and handling editors have recommended publication, but also suggest some minor revisions to your manuscript. Therefore, I invite you to respond to the comments and revise your manuscript.

Because the schedule for publication is very tight, it is a condition of publication that you submit the revised version of your manuscript before 04-Jul-2019. Please note that the revision deadline will expire at 00.00am on this date. If you do not think you will be able to meet this date please let me know immediately.

Best wishes,
Dr Laura Smith
Publishing Editor, Journals

Royal Society of Chemistry
Thomas Graham House

Science Park, Milton Road
 Cambridge, CB4 0WF
 Royal Society Open Science - Chemistry Editorial Office

On behalf of the Subject Editor Professor Anthony Stace and the Associate Editor Dr Andrew Harned.

RSC Associate Editor:

Comments to the Author:

The reviewers are generally satisfied by the responses provided by the authors, but would like to see these responses incorporated into the manuscripts itself. I very much agree with this sentiment as it will increase transparency and openness in the scientific record. In addition, the reviewers offer a few minor suggested changes that should be included in a revised manuscript.

RSC Subject Editor:

Comments to the Author:

(There are no comments.)

Reviewer comments to Author:

Reviewer: 2

Comments to the Author(s)

The revised manuscript addressed most of my previous concerns. However, the authors should address the following before consideration of publishing this work. In particular, the answers to many questions should be included in the text to increase clarification rather than simply being used to answers the reviewer's questions.

For the Q3, the authors' answer is relevant. The following paragraph should be embedded within the text somewhere.

"Resveratrol shows extremely strong effects on suppression of tubulin assembly through interaction with the colchicine binding site of tubulin, resulting in extensive inhibition of cell growth and angiogenesis. However, it is difficult to be made pharmaceutical preparation due to low polarity. Thus, we designed and synthesized a series of novel small-molecule derivatives of resveratrol by adding a carboxyl group on the carbon-carbon double bond for a better water solubility as potential anti-cancer agent. Electron-withdrawing and electron-donating group were chosen as R1-R8 substituents to get variety compounds for SAR studies."

For the Q4, I don't feel that the authors understood the meaning of "rationalize", although they answered the question indirectly by saying "A549, Hela and HepG2 cell lines are representative". The detailed description of these cell lines should be moved to the experimental session. "A549, Hela and HepG2 cell lines are representative" should be included in the main text to tell readers the reason to choose these cell lines.

Foe the Q5, thanks for providing the answers. This answer should be included in the text rather than just letting me know.

For the Q6, Q7, the answers should be included in the text.

For the Q8, the format to present "mean +/- error" was not correct. For instance, "50.21±3.85"

should be "50±4"; "28.17±0.53" should be "28.2±0.5" for statistical reasons. The changes should be made via the whole manuscript.

For the Q9, the current modeling could not present the accuracy to 0.01 A and 0.01 degree. For instance, "2.48" is better shown as "2.4"; "114.07" is better shown as "141". These numbers should be updated.

Reviewer: 1

Comments to the Author(s)

The authors have addressed my revisions well. I recommend publication upon correction of the revision noted below.

Figure 1. The bond angle of the COOH on the double bond needs to be corrected.

Author's Response to Decision Letter for (RSOS-190125.R1)

See Appendix B.

Decision letter (RSOS-190125.R2)

16-Jul-2019

Dear Dr Jiang:

Title: Design, synthesis and biological evaluation of a series of new resveratrol analogues as potential anti-cancer agents
Manuscript ID: RSOS-190125.R2

Thank you for submitting the above manuscript to Royal Society Open Science. On behalf of the Editors and the Royal Society of Chemistry, I am pleased to inform you that your manuscript will be accepted for publication in Royal Society Open Science subject to minor revision in accordance with the referee suggestions. Please find the reviewers' comments at the end of this email.

The reviewers and handling editors have recommended publication, but also suggest some minor revisions to your manuscript. Therefore, I invite you to respond to the comments and revise your manuscript.

Because the schedule for publication is very tight, it is a condition of publication that you submit the revised version of your manuscript before 25-Jul-2019. Please note that the revision deadline will expire at 00.00am on this date. If you do not think you will be able to meet this date please let me know immediately.

To revise your manuscript, log into <https://mc.manuscriptcentral.com/rsos> and enter your Author Centre, where you will find your manuscript title listed under "Manuscripts with

Decisions". Under "Actions," click on "Create a Revision." You will be unable to make your revisions on the originally submitted version of the manuscript. Instead, revise your manuscript and upload a new version through your Author Centre.

Best wishes,

Dr Laura Smith
Publishing Editor, Journals

On behalf of the Subject Editor Professor Anthony Stace and the Associate Editor Dr Andrew Harned.

RSC Associate Editor

Comments to the Author:

The authors have now incorporated the responses as requested by the reviewers. After reading through the manuscript again, I have noted a few additional minor corrections that need to be made.

(1) Table 3: Please recheck the angle for the H-bond and correct as needed. I think the reviewer may have had a typo in their suggestion.

(2) "equ" is not an appropriate abbreviation to "equivalent". Please change all instances to "equivalent" or "equivalents" as appropriate.

(3) The identities of compounds a-v are still not defined in Scheme 1. Only the substituents explored at each position are given. Perhaps this can be addressed by adding the general structure of the compounds above Table 1.

(4) Provide structures with each NMR spectrum

(5) Experimental section: provide details of column chromatography and preparative HPLC

Reviewer comments to Author:

Author's Response to Decision Letter for (RSOS-190125.R2)

See Appendix C.

Decision letter (RSOS-190125.R3)

05-Aug-2019

Dear Dr Jiang:

Title: Design, synthesis and biological evaluation of a series of new resveratrol analogues as potential anti-cancer agents

Manuscript ID: RSOS-190125.R3

It is a pleasure to accept your manuscript in its current form for publication in Royal Society Open Science. The chemistry content of Royal Society Open Science is published in collaboration with the Royal Society of Chemistry.

Yours sincerely,
Dr Ellis Wilde
Publishing Editor, Journals

On behalf of the Subject Editor Professor Anthony Stace and the Associate Editor Dr Andrew Harned.

RSC Associate Editor
Comments to the Author:
(There are no comments.)

Reviewer(s)' Comments to Author:

Appendix A

Dear Dr Laura Smith,

Thank you very much for giving us good advice for the manuscript.

We have revised the manuscript according to the Reviewers' comments. Spelling and grammar of the manuscript had been addressed by an editor company. The following is a point-to-point response to the Reviewers' comments. Thank you very much for all your help and looking forward to hearing from you soon.

Best,

Jiang

Reviewers' Comments to Author:

Reviewer: 1

Comments to the Author(s)

The authors present the synthesis and biological activity assays of a series of novel resveratrol analogues. The study identifies compound r as the most potent. However it is considerably less potent than Taxol. There is not data provided as to the purity of these compounds, no HPLC, no NMR traces. This, combined with the issues listed below prevent me from supporting publication at this time.

Major issues:

No NMR, No HPLC data. No purity information Is the biological activity attributable to the compound or an impurity?

Response: We added NMR in the supporting materials. Purity was monitored based on the value of the melting point and NMR.

In the morphological observations it is not clear why the compounds were combined with cisplatin. It is not stated in the text that cisplatin was used but it was in the figure legend.

Response: In the morphological observations, cisplatin was used as positive control drugs. Statement had been added in the text.

The NMR data is assigned to three significant figures. How accurate is this?

Response: The NMR data had been assigned to two significant figures.

Minor issue:

The manuscript needs further proof reading to fix spelling and grammar issues.

For example the use of N₂ for protection is not a clear description of what I assume to mean inert atmosphere.

'Lead compounds are often used as lead compounds for drug discovery' This is not correct. They are often used as hit compounds but never leads.

Response: Spelling and grammar of the manuscript had been addressed by an editor company.

Figure 5, the graph is too small to read the labels.

Response: Figure 5 had been amplified in this manuscript.

Reviewer: 2

Comments to the Author(s)

In this manuscript, Yang et. al. reported the synthesis of 22 resveratrol derivatives and characterized their anti-cancer activities. The synthesis is straightforward by coupling two scaffolds under a single condition. The resulting compounds were examined with MTT assay and, with the compound “r”, for its effects on cellular morphology, tubulin formation, apoptosis and cell cycle. The authors claimed tubulin as a potent target of the compound “r” based on the evidence of molecular docking. While the exact mechanism of action for the compound remains unclear, the decent effort has been made to prepare these compounds and to characterize the most promising one. I suggest publishing this work after addressing the following concerns:

(1) There are numerous grammar errors via the text. A few examples are: Line 10, missing “of” between “series” and “new”; Line 31, “agent” should be “agents”; Line 33, “cell cycle” should be “cell cycle assays”; Line 34, should be “... arrested cell cycle at S-phase; Line 35, the sentence “... compound r anti-cancer drug targeting tubulin...” is not clear; Line 35, “were” should be “was”; Line 36, “the tubulin active pocket” should be “the active pocket of tubulin”; Line 36, “probable” should be “possible”. There are more via the text. I suggest authors to seek the help of an editor company or a native speaker to address these issues.

Response. Spelling and grammar of the manuscript had been addressed by an editor company.

(2) For Scheme 1, the substituents of R1-R8 should be listed given that there are only 22 compounds.

Response: The substituents of R1-R8 had been added in Scheme 1 of the manuscript.

(3) For the section 2.1, the authors should provide clear rationale why this scaffold was chosen. What is the reason for selecting R1-R8 substituents in the context of prior knowledge of the family of compounds.

Response. Resveratrol shows extremely strong effects on suppression of tubulin assembly through interaction with the colchicine binding site of tubulin, resulting in extensive inhibition of cell growth and angiogenesis. However, it is difficult to be made pharmaceutical preparation due to low polarity. Thus, we designed and synthesized a series of novel small-molecule derivatives of resveratrol by adding a carboxyl group on the carbon-carbon double bond for a better water solubility as potential anti-cancer agent. Electron-withdrawing and electron-donating group were chosen as R1-R8 substituents to get variety compounds for SAR studies.

(4) For 2.2.1, the choices of cell lines should be rationalized in the context of known biology.

Response: A549, Hela and HepG2 cell lines were purchased from the Shanghai Institute of Cell Biology (Shanghai, China). The Normal african green monkey kidney Vero cells were a gift from Dr. xiumiao He at Guangxi University for Nationalities. LO2 cells were a gift from Dr. xing Lin at Guangxi Medical University. The HepG2, Hela, LO2 and Vero cells were cultured in Dulbecco’s modified Eagle’s medium (DMEM, Gibco) and the A549 cells were cultured in RPMI1640 (Gibco), both media were supplemented with 10% fetal calf serum(PAA, Austria) and antibiotics

(100 IU/ml penicillin, 100 IU/ml streptomycin; Amresco). All cell cultures were maintained at 37 °C in a humidified atmosphere supplemented with 5% CO₂. A549, HeLa and HepG2 cell lines are representative. All compounds were investigated their anti-proliferative activities and cytotoxicity.

(5) Why was Vero cells chosen as control?

Response: Vero cells are non-tumorigenic cell lines and are important raw materials as the research and production of biological products. Vero cells show low tumorigenicity and had been approved for use in the production of human virus vaccines. Thus, Vero cells was chosen as control cytotoxicity. In addition, all compounds were evaluated for their toxicity against normal human hepatocytes cells line (LO2) using MTT assays.

(6) Need the rationale the experiment 2.2.3. Why does the morphology matter?

Response: Morphological observations directly reflect the effects of drugs on cells.

(7) There is also a limitation of docking. In addition, even the compound **r** indeed targets tubulin. The authors still could not conclude that the biology of the compound is associated with tubulin interaction but not other unknown targets. The limitation should be clearly spoken out.

Response: It's reported that resveratrol derivatives shows extremely strong effects on suppression of tubulin. Compound **r** was docked into the paclitaxel binding site in tubulin to provide a possible target. Whether compound **r** is associated with other unknown targets under studying. Compound **r** was evaluated as β -tubulin inhibitors by immunofluorescence microscopy assay. Compared with paclitaxel, cells treated with compound **r** exhibited similar microtubule architecture, irregular microtubule networks and nuclear localization.

(8) There is no error range of IC₅₀ in Table 1 and CC₅₀ in Table. In addition, the accuracy could not reach the current digits. These data should be represented including the correct accuracy and errors.

Response: Values in Table 1 are averages of three independent determinations. The correct accuracy and errors of these data had been added in this manuscript.

(9) For Table 3, the distances and angles with such accuracy could not be obtained with the current modeling methods. The data need to be presented in the correct format.

Response: The data had been assigned to two significant figures.

Appendix B

Dear Dr Laura Smith

Thank you very much for your comments and suggestions.

We have revised the manuscript according to the Reviewers' comments. The following is a point-to-point response to the two Reviewers' comments. Thank you very much for all your help and looking forward to hearing from you soon.

Best,

Jiang

Reviewer comments to Author:

Reviewer: 2

Comments to the Author(s)

The revised manuscript addressed most of my previous concerns. However, the authors should address the following before consideration of publishing this work. In particular, the answers to many questions should be included in the text to increase clarification rather than simply being used to answers the reviewer's questions.

For the Q3, the authors' answer is relevant. The following paragraph should be embedded within the text somewhere.

“Resveratrol shows extremely strong effects on suppression of tubulin assembly through interaction with the colchicine binding site of tubulin, resulting in extensive inhibition of cell growth and angiogenesis. However, it is difficult to be made pharmaceutical preparation due to low polarity. Thus, we designed and synthesized a series of novel small-molecule derivatives of resveratrol by adding a carboxyl group on the carbon-carbon double bond for a better water solubility as potential anti-cancer agent. Electron-withdrawing and electron-donating group were chosen as R1-R8 substituents to get variety compounds for SAR studies.”

Response: This paragraph has been embedded within the text .

For the Q4, I don't feel that the authors understood the meaning of “rationalize”, although they answered the question indirectly by saying “A549, Hela and HepG2 cell lines are representative’. The detailed description of these cell lines should be moved to the experimental session. “A549, Hela and HepG2 cell lines are representative’ should be included in the main text to tell readers the reason to choose these cell lines.

Response: The detailed description of these cell lines has been moved to the experimental session.

Foe the Q5, thanks for providing the answers. This answer should be included in the text rather than just letting me know.

Response: The answers have been embedded within the text .

For the Q6, Q7, the answers should be included in the text.

Response: The answers have been embedded within the text .

For the Q8, the format to present “mean +/- error” was not correct. For instance, “50.21±3.85” should be “50±4”; “28.17±0.53” should be “28.2±0.5” for statistical reasons. The changes should be made via the whole manuscript.

Response: The format to present “mean +/- error” had been revised in this manuscript.

For the Q9, the current modeling could not present the accuracy to 0.01 A and 0.01 degree. For instance, “2.48” is better shown as “2.4”; “114.07” is better shown as “141”. These numbers should be updated.

Response: These numbers have been updated.

Reviewer: 1

Comments to the Author(s)

The authors have addressed my revisions well. I recommend publication upon correction of the revision noted below.

Figure 1. The bond angle of the COOH on the double bond needs to be corrected.

Response: The bond angle of the COOH on the double bond has been corrected.

Appendix C

Dear Dr Laura Smith.

Thank you very much for your comments and suggestions.

We have revised the manuscript according to your comments. The following is a point-to-point response to your comments. Thank you very much for all your help and looking forward to hearing from you soon.

Best,

Jiang

RSC Associate Editor

Comments to the Author:

The authors have now incorporated the responses as requested by the reviewers. After reading through the manuscript again, I have noted a few additional minor corrections that need to be made.

(1) Table 3: Please recheck the angle for the H-bond and correct as needed. I think the reviewer may have had a typo in their suggestion.

Response: We had invited researcher who do molecular docking to rechecked the angle for the H-bond and found no problem.

(2) "equ" is not an appropriate abbreviation to "equivalent". Please change all instances to "equivalent" or "equivalents" as appropriate.

Response: "equ" had been changed into "equivalent" or "equivalents".

(3) The identities of compounds a-v are still not defined in Scheme 1. Only the substituents explored at each position are given. Perhaps this can be addressed by adding the general structure of the compounds above Table 1.

Response: The general structure of the compounds had been added in **Table 1** of the manuscript.

(4) Provide structures with each NMR spectrum

Response: Structures of compounds had been added in each ¹H-NMR spectrum.

(5) Experimental section: provide details of column chromatography and preparative HPLC

Response: The details of column chromatography and preparative HPLC have been provided in experimental section.